# Etiologies of Zoonotic Tropical Febrile Illnesses That Are Not Part of the Notifiable Diseases in Colombia

**DOI:** 10.3390/microorganisms11092154

**Published:** 2023-08-25

**Authors:** Carlos Ramiro Silva-Ramos, Álvaro A. Faccini-Martínez, Cristian C. Serna-Rivera, Salim Mattar, Marylin Hidalgo

**Affiliations:** 1Grupo de Enfermedades Infecciosas, Departamento de Microbiología, Facultad de Ciencias, Pontificia Universidad Javeriana, Bogotá 110231, Colombia; cramiro-silva@javeriana.edu.co; 2Servicio de Infectología, Hospital Militar Central, Bogotá 110110, Colombia; afaccini@gmail.com; 3Servicios y Asesorías en Infectología—SAI, Bogotá 110110, Colombia; 4Grupo de Investigación en Ciencias Veterinarias (CENTAURO), Línea de Investigación Zoonosis Emergentes y Re-Emergentes, Facultad de Ciencias Agrarias, Universidad de Antioquia, Medellín 050034, Colombia; cristian.sernar@udea.edu.co; 5Grupo de Investigación en Genética, Biodiversidad y Manejo de Ecosistemas (GEBIOME), Departamento de Ciencias Biológicas, Facultad de Ciencias Exactas y Naturales, Universidad de Caldas, Manizales 170004, Colombia; 6Instituto de Investigaciones Biológicas del Trópico, Universidad de Córdoba, Montería 230001, Colombia; smattar@correo.unicordoba.edu.co

**Keywords:** *Alphavirus*, *Anaplasma*, arbovirus, arenavirus, *Bartonella*, *Borrelia*, Colombia, *Coxiella burnetii*, *Ehrlichia*, hantavirus, *Orientia*, *Orthobunyavirus*, *Rickettsia*

## Abstract

In Colombia, tropical febrile illnesses represent one of the most important causes of clinical attention. Febrile illnesses in the tropics are mainly zoonotic and have a broad etiology. The Colombian surveillance system monitors some notifiable diseases. However, several etiologies are not monitored by this system. In the present review, we describe eleven different etiologies of zoonotic tropical febrile illnesses that are not monitored by the Colombian surveillance system but have scientific, historical, and contemporary data that confirm or suggest their presence in different regions of the country: *Anaplasma*, Arenavirus, *Bartonella*, relapsing fever group *Borrelia*, *Coxiella burnetii*, *Ehrlichia*, Hantavirus, Mayaro virus, * Orientia*, Oropouche virus, and *Rickettsia*. These could generate a risk for the local population, travelers, and immigrants, due to which they should be included in the mandatory notification system, considering their importance for Colombian public health.

## 1. Introduction

Colombia is a Latin American country whose climate is isothermal; it has a constant temperature with only rainy and dry seasons with slight variation throughout the year. It has five natural geographic regions: the Amazon Rainforest, the Andes Mountain Range region, the Caribbean coast, the Grassland Plains, and the Pacific coast. At least 85% of the national territory is composed of tropical and subtropical regions; thus, tropical diseases represent one of the most critical affections for Colombians and travelers who visit the country [1,2,3].

Febrile illnesses represent the leading cause of clinical attention in tropical and subtropical regions [4]. The etiology of tropical fevers is broad and includes a significant number of pathogens, most of them zoonotic, which may have arisen from wild animal species favored by different wildlife intervention activities, such as intensified agriculture, accelerated urbanization, and eco-tourism, among others [5,6,7].

Some etiologies of zoonotic tropical febrile illnesses (e.g., leptospirosis, yellow fever, and equine encephalitis) are notifiable diseases for the Colombian national surveillance system (Instituto Nacional de Salud (INS)—https://www.ins.gov.co accessed on 1 May 2023). However, several others, whose importance has been demonstrated in other countries (e.g., spotted fever group rickettsioses in Brazil and scrub typhus in Chile) [8,9], are not monitored by local authorities despite the scientific, historical, or contemporary evidence that confirms or suggests their circulation in Colombia.

Although for some of these non-notifiable zoonotic pathogens (e.g., *Rickettsia*), the national surveillance center (INS) can perform a confirmatory diagnosis in case of high clinical suspicion, diagnostic methods for others are only available in some higher education research groups’ laboratories that work with these zoonotic infectious agents, using commercial kits or in-house methods. Reference laboratories from international centers such as the CDC (Centers for Disease Control and Prevention) are an alternative way to reach a confirmatory diagnosis, mainly in case of outbreaks of an unknown disease [10]. In addition, research groups have contributed to knowledge about these non-notifiable zoonotic febrile illnesses and their repercussions on the local population’s health.

This review aims to summarize and describe the evidence of other etiologies of zoonotic tropical fevers that are, or probably are, circulating in the country and are not part of the notifiable diseases in Colombia.

## 2. Arenavirus

Mammarenaviruses (Arenaviridae family) have traditionally been divided into two distinct groups: Old World mammarenaviruses, which include viruses from Africa (e.g., Lassa virus (LASV) and Mopeia virus) and the worldwide distributed Lymphocytic Choriomeningitis virus (LCMV); and the New World mammarenaviruses, which include viruses from the Americas (e.g., Pichindé virus and Tacaribe virus) [11]. Human pathogenic mammarenaviruses include LCMV, Junin virus, Machupo virus, LASV, Guanarito virus (GTOV), Sabiá virus, Chapare virus, and Lujo virus [12]. Overall, arenavirus infections can be mild or severe, and they usually develop as febrile illnesses with a history of exposure to rodents or their excreta, which can be helpful for early suspicion of the disease [13].

LCMV infection is typically a biphasic disease; first, it develops as a non-specific febrile illness that lasts from a few days to almost three weeks, often followed by a second phase in which the central nervous system gets affected by aseptic meningitis, or encephalitis in severe cases, which lasts from 1 to 4 weeks and mostly resolves without sequelae [14]. In cases of arenaviral hemorrhagic fevers caused by New World mammarenaviruses, the progression of the disease may be faster than in other arenaviral infections (e.g., LASV infection and LCMV infection). Arenaviral hemorrhagic fevers usually present as an acute undifferentiated febrile illness (AUFI) associated with oral mucous hemorrhage and bleeding gums. Three to four days after the onset of symptoms, severe prostration can occur, and severe hemorrhagic signs can develop (e.g., melena and petechiae); white blood cell and platelet counts are usually low, and neurologic involvement associated with convulsions can occur [15].

Laboratory diagnosis includes the detection of IgM or IgG antibodies with enzyme-linked immunoassays (ELISAs) or indirect fluorescent antibody tests (IFA); the detection of the viral genome with polymerase chain reaction (PCR); or viral isolation from serum, blood, or cerebrospinal fluid samples. However, the last method is rarely performed due to the biosafety risk; the need for Biosafety Level 4 laboratories; and the fact that it is resource intensive, when other techniques, such as PCR, are much easier to perform [16,17,18].

In Colombia, Pichindé virus, an arenavirus whose reservoirs are *Nephelomys albig-ularis* (previously called *Oryzomys albigularis*) rodents, is actively circulating, and apparently, it is not associated with human infection [19,20]. However, serological studies performed in three regions of Antioquia, Cesar, and Córdoba departments have evidenced seropositivity to mammarenaviruses among different population groups (Table 1) [21,22,23], as well as one possible case of arenaviral infection in which seroconversion was evidenced (Figure 1A) [21]. These findings suggest that another arenaviruses may be present in Colombia, which apparently can be infective for humans.

In addition, the circulation of GTOV, the etiological agent of Venezuelan hemorrhagic fever (VHF), was already confirmed with plaque reduction neutralization testing (PRNT) in *Zygodontomys brevicauda* rodents from rural areas on the Caribbean coast of the department of Córdoba [34]. Although no human cases of VHF have been reported in Colombia to date, the proximity of the Colombian and Venezuelan eastern plains, their typical features (climate, the presence of small mammals, socioeconomic conditions, and land use), and the increasing flow of Venezuelan migrants and refugees due to the complex humanitarian crises in the neighboring country suggest that VHF, which is endemic in the Venezuelan western states of Portuguesa and Barinas [35,36], might be present in Colombia and may be one of the causes of AUFIs in the Colombian–Venezuelan border region [36]. LCMV has also been detected in *Mus musculus* rodents collected from Sincelejo, Sucre [37]; nevertheless, no human case due to this agent was reported.

## 3. Hantavirus

Orthohantaviruses (Hantaviridae family) are emerging viruses whose reservoirs are small rodent species and affect approximately 150,000 to 200,000 people worldwide annually [38]. Hantavirus infection occurs after exposure to contaminated aerosols or excreta from infected rodents [38]. A total of 40 hantavirus species have been described, 22 of which are pathogenic for humans. Those limited to Europe and Asia are known as Old World hantaviruses (e.g., Puumala virus and Hantaan virus), and those reported in America are known as New World hantaviruses (e.g., Sin Nombre virus and Andes virus) [38]. Clinically, two forms of hantavirus disease have been described: hemorrhagic fever with renal syndrome (HFRS) and hantavirus cardiopulmonary syndrome (HCPS) [39,40].

HFRS is caused by Old World hantaviruses. It is a non-specific febrile illness with signs of renal damage such as oliguria, proteinuria, and hematuria in severe cases. Low platelet counts and lymphocytosis are the main laboratory abnormalities. The mortality rate ranges from less than 1% to 15% and depends on the type of virus implicated, with Hantaan virus being the most pathogenic [39,41].

On the other hand, HCPS is caused by New World hantaviruses. It is a severe pulmonary acute illness that clinically develops as a non-specific febrile illness that evolves as classic acute respiratory distress syndrome. It has a mortality rate of more than 40% due to pulmonary edema, respiratory insufficiency, or cardiogenic shock, and is fatal than HFRS [39,42].

Laboratory diagnosis of hantavirus infection is usually made using serological methods, which can include the detection of IgM antibodies with ELISA or evidence of seroconversion of IgG antibodies between acute and convalescent serum samples. PCR can detect the presence of the viral RNA, which can be considered a confirmatory diagnosis. In addition, hantavirus antigens in formalin-fixed tissues can be evidenced with immunohistochemistry (IHC), which can be used in post-mortem cases to establish the cause of death [43]. Although the virus can be isolated, this process must be only performed in reference laboratories [43].

In the Americas, since 1993, when the first case of HCPS was identified in the United States, countries such as Brazil, Argentina, and Chile, among others, have continuously been describing novel HCPS human cases [44]. However, some confirmed cases with mild clinical courses, incompatible with the classic HCPS, have also been reported in these countries [45,46]. In Colombia, the first record of human exposure to hantavirus was reported in 2004 among rural male workers from 12 towns of Córdoba and Sucre departments [26]. Ten years later, in Monteria (Córdoba department), the first confirmed case of hantavirus disease, not clinically compatible with HCPS, was identified and diagnosed using seroconversion [47]. Further studies performed in the Caribbean (Córdoba department) and Orinoquia regions (Meta department) identified novel cases due to hantavirus infection among febrile patients, finding 6% (6/100) and 3% (3/100) of cases, respectively, all of them confirmed using seroconversion with IgG ELISA [27,48]. In neither study, cases were clinically compatible HCPS, raising the possibility that other hantaviruses related with mild cases of HFRS (e.g., Seoul virus) might be circulating in the country [49].

Furthermore, several serological studies performed in different Colombian regions have evidenced human exposure to hantaviruses among different population groups, such as healthy inhabitants from six urban municipalities of Córdoba department [24], febrile patients in urban settlements of the Urabá region (Antioquia department) in whom no seroconversion was evidenced [21], and indigenous populations from rural communities of Córdoba (Embera Katío and Tuchín communities) and Cesar departments (Kankuamos community) [22,23,25], suggesting that at least one hantavirus native to Colombia is circulating among rural inhabitants and that, probably, another hantavirus might be present in urban regions (Table 1) (Figure 1B). Exposure to hantaviruses has also been evidenced in several rodent species in the departments of Córdoba, Sucre, and Antioquia [50,51,52], as well as the description of a novel hantavirus designated as “Necoclí” virus [53], whose importance for human and animal health is still unknown. This evidence suggests that at least one human pathogenic hantavirus is present in Colombia, about which little is known to date. Further studies among wild rodent populations are thus needed because some species may be acting as reservoirs and shedders of these viruses.

## 4. Mayaro Virus

Mayaro virus (MAYV) is an arbovirus that belongs to the *Alphavirus* genus (Togaviridae family). It is part of the Semliki complex along with other viruses, such as Chikungunya, Semliki Forest, Ross River, and O’nyong-nyong [54]. In its natural wild cycle, MAYV is transmitted by female mosquitoes of the genus *Haemagogus*, mainly *Haemagogus janthinomys*; however, its introduction into urban environments also involves other mosquito species, such as *Aedes* and *Culex* spp., which may be acting as vectors of MAYV [55,56].

Cases due to MAYV have been limited to several regions of Central and South America, mainly those surrounding the Amazon basin. Over the last few years, MAYV cases have continuously increased, suggesting that it could be imperative to identify it as a novel epidemic arbovirus [57]. Due to the clinical, epidemiological, and biological similarities with the Chikungunya virus (CHIKV), which is also present in the same regions, it may be probable that cases due to MAYV are misdiagnosed as CHIKV [58].

MAYV usually causes a mild and self-limited AUFI with unspecific clinical manifestations, of which arthralgia and maculopapular rash are the most important during its acute phase; fever during MAYV infection is usually biphasic, reappearing after an afebrile period, which may differentiate it from other arboviral etiologies [54,59]. During the convalescent phase, more than half of the patients suffer from long-term arthralgia, persisting for several months, similar to CHIKV infection [60]. Although most cases are mild and self-limited, MAYV infection can also produce several complications, such as neurological compromise, heart damage, chronic intermittent fever, and hemorrhages, which can be so severe that death can occur [54].

Laboratory diagnosis can be performed using virological methods to detect viral RNA with PCR or cell culture isolation. The latter is rarely used as a first-line diagnostic tool and is only performed in reference laboratories [61,62]. Serological methods can also be used; ELISA or other immunoassay methods can detect anti-MAYV IgM antibodies. However, results must be interpreted carefully, as cross-reactivity can occur with other alphaviruses. Thus, its detection is only presumptive, and confirmation requires demonstrating IgG seroconversion in paired serum samples without evidence of seroconversion to other alphaviruses [63,64].

In Colombia, although no clinical case due to MAYV has been reported to date, the circulation of this virus in several regions of the country has already been evidenced. The first report came in 1961, when MAYV was successfully isolated from anthropophilic *Psorophora* mosquitoes collected in San Vicente de Chucurí, Santander department [65], raising the possibility that these arthropods may be involved in MAYV local eco-epidemiology. In 1964, seropositivity was found among the rural populations from several regions of Chocó, Tolima, Cundinamarca, Santander, Meta, and Vichada departments, finding different percentages of exposure, with the highest ones being 60% (3/5) and 24.5% (38/155), and being found in Cumaribo municipality (Vichada department) and Barrancabermeja municipality (Santander department), respectively [30]. In 1969, seropositivity was also found among military recruits from different regions of the country [32]. In 1970, seropositivity was found among indigenous populations in the region of Araracuara, Puerto Santander, Amazonas department [28]. Since then, an epidemiological silence took place until 2007, when during a febrile illness sentinel program, seropositivity was detected in Guaviare and Magdalena departments [31]. Furthermore, more recently, in 2022, during an arboviral surveillance study performed in the department of Cauca, seropositivity was found among healthy individuals (Table 1) (Figure 1C) [29]. This evidence suggests that MAYV is actively circulating in rural regions of the country, apparently generating more asymptomatic cases.

Although its importance may be underestimated to date, a study showed, using mathematical analysis, that several regions are at greater risk of a possible MAYV epidemic once the mosquitoes adapt to an urban environment or MAYV adapts to urban or urban–wild transition mosquitoes such as *Aedes aegypti* or *Aedes albopictus*, respectively [66]. Thus, MAYV might be one of the most critical threats in the future due to accelerated urbanization and deforestation, adapting to other anthropophilic mosquitoes such as *Aedes* spp., whose competence as a MAYV vector has already been demonstrated and are circulating in an urban cycle [67].

## 5. Oropouche Virus

Oropouche virus (OROV) is a member of the Simbu serogroup of the *Orthobunyavirus* genus (Bunyaviridae family); it is the etiological agent of Oropouche fever, one of the etiologies of AUFIs, mainly in the Amazon basin, and probably a candidate for the next great arboviral epidemic across the Americas, along with Venezuelan equine encephalitis virus and MAYV [68,69]. OROV is a vector-borne infectious agent mainly transmitted by the bite of *Culicoides paraensis* [70]. However, other mosquito species, such as *Culex quinquefasciatus*, *Coquilletidia venezuelensis*, and *Aedes serratus*, might also be involved, with *C. paraensis* and *Cx. quinquefasciatus* being the vectors related to the urban cycle and the remaining two species being in their natural wild cycle [69]. Some mammals and wild bird species might act as natural reservoirs in the natural wild cycle [71,72].

The first OROV human infection was notified in the village of Vega de Oropouche, Trinidad and Tobago [73]. Since then, isolated cases have been reported in many South American countries (e.g., Argentina, Bolivia, and Ecuador), and disease outbreaks have been reported in Brazil, Peru, and Panama [69].

Clinically, Oropouche fever develops as an AUFI with non-specific symptomatology, such as headache, myalgia, arthralgia, chills, and dizziness [72,74]. In more than half of cases, clinical manifestations recur a few days after the initial febrile episode. Even though severe cases are unusual, complications can occur, of which aseptic meningitis is the most significant [75]. Recovery is complete without any apparent sequel, even in severe cases; to date, no fatal cases have been reported [74].

Laboratory diagnosis can be made using serological procedures in order to detect specific IgG and IgM antibodies with ELISA, IFA, PRNT, or Hemagglutination Inhibition (HAI) tests [72]. A single positive serum result in a clinically compatible patient can highly suggest OROV infection; however, confirmatory diagnosis requires the isolation of the virus, which is only performed in reference laboratories, or the detection of viral RNA with PCR [72].

In Colombia, the first record of the presence of OROV was reported in 1961, in which seropositivity was found among 27.3% (6/22) of primates sampled in the municipality of Lizama, Santander department, using Neutralization Test 50 LD50 [30]. No human cases were reported until 2021, when a case of Oropouche fever was reported in the municipality of Turbaco (Bolivar department) and was confirmed with viral culture, IFA, and PCR [76], with this being the first clinical case of Oropouche fever confirmed in Colombia.

Only two additional studies have been performed in which seropositivity was found among different populations from Colombia, including patients from Guaviare department enrolled in a febrile illness sentinel program [31] and healthy rural villagers from four municipalities of Cauca department [29]. In addition, one study performed among febrile patients in four urban municipalities of Amazonas, Meta, Norte de Santander, and Valle Del Cauca departments detected the presence of OROV using molecular methods (Table 1) (Figure 1D) [33]. These findings suggest that local transmission of OROV infection is occurring in Colombia, probably more frequently in rainy seasons, and that Oropouche fever is one of the causes of AUFIs in the country. However, its importance remains underestimated due to the absence of studies and previous reports in many regions throughout the country.

## 6. *Anaplasma*

*Anaplasma* is a genus of obligate intracellular Gram-negative bacteria of veterinary and human health importance transmitted via tick bite [77]. Two *Anaplasma* species, *Anaplasma phagocytophilum* and *Anaplasma capra*, have been implicated as human pathogens. However, it has been suggested that anaplasmosis may be a significant emerging infectious disease in the future, as several novel and yet unrecognized *Anaplasma* species are continuously being detected in a wide range of hosts [78,79].

The main human pathogenic species is *A. phagocytophilum*, the etiological agent of human granulocytic anaplasmosis (HGA), a tick-borne infectious disease transmitted by several *Ixodes* species [80]. Clinically, it can range from asymptomatic infection to fatal disease. It usually develops one or two weeks after tick exposure as a non-specific febrile illness accompanied by severe headache, myalgia, arthralgia, sweating, and stiff neck as the most common symptoms [80]. The disease can be severe in 35% of cases; however, the fatality rate is less than 1%, and life-threatening complications include acute respiratory distress syndrome, acute renal failure, and hemodynamic collapse [81,82].

Laboratory confirmatory diagnosis includes PCR on DNA extracted from blood samples, IgG seroconversion with IFA, and direct observation of morulae (microcolonies of *Anaplasma*) in the cytoplasm of granulocytes of peripheral blood smears with microscopy. Confirmatory diagnosis should be performed in clinically compatible patients with risk factors such as living or having traveled to endemic regions or those bitten by a tick [83].

Although the tick species recognized as competent vectors of *A. phagocytophilum* are not present within Colombia up to now, one probable case of anaplasmosis has been reported in a febrile livestock farmer in the city of Cartagena (Atlántico department), whose diagnosis was only performed using microscopy with Wright–Giemsa staining and who fully recovered after treatment with doxycycline [84]. Another probable case of co-infection of human anaplasmosis/dengue fever was reported in the municipality of Villeta (Cundinamarca department) [85]. Although no recognized *Anaplasma* vectors have been found in Colombia, studies performed in Argentina have reported the detection of *Anaplasma* spp. of unknown pathogenicity in anthropophilic ticks of *Amblyomma* spp. [86,87], suggesting another competent vector may be present in South America.

Exposure to *Anaplasma* spp. has also been evidenced in different Colombian regions, such as livestock farming workers in Córdoba and Sucre departments [88], febrile patients from Villeta municipality (Cundinamarca department) [85], livestock farming workers from San Pedro de Los Milagros (Antioquia department) [89], and febrile patients from the Magdalena Medio region (Table 2) (Figure 1E) [90]. These findings suggest and reinforce the possibility that at least one human pathogenic *Anaplasma* sp. may be present in the country.

## 7. *Bartonella*

*Bartonella* is a group of fastidious intra-erythrocytic Gram-negative bacteria. They have many animal reservoirs, including humans, and are transmitted by a wide range of vectors, including sand flies, lice, fleas, and probably ticks [100,101]. *Bartonella bacilliformis*, *Bartonella quintana*, and *Bartonella henselae* are classically known as the pathogenic species of this genus. Infections caused by *Bartonella* spp. can be mild and sometimes asymptomatic. However, there are also clinically symptomatic cases, with some of them being severe and even fatal depending on the host (e.g., immune status and underlying diseases) and microorganism factors (e.g., species involved and bacterial load) [100,102].

The most known diseases due to *Bartonella* spp. include Carrion’s disease, trench fever, and cat-scratch disease. However, other diseases include chronic lymphadenopathy, bacteremia, culture-negative endocarditis, bacillary angiomatosis, and bacillary peliosis [102]. Carrion’s disease is a biphasic illness caused by *B. bacilliformis* and transmitted by sand flies of *Lutzomya* spp. During its acute phase, the disease is known as Oroya fever. This acute febrile hemolytic illness causes anemia and non-specific symptomatology, with a fatality rate as high as 90% if patients are not opportunely treated. The chronic phase of the disease is known as “verruga peruana” (Peruvian wart), which is a blood-filled nodular skin lesion similar to a hemangioma that can persist for several months [103]. Trench fever is caused by *B. quintana* and is transmitted by the clothing louse *Pediculus humanus*. It can develop as a mild self-limited febrile illness or a prolonged recurrent debilitating febrile illness with asymptomatic periods in between [104]. *B. henselae* mainly causes cat-scratch disease, although other species, such as *Bartonella clarridgeiae*, can also be implicated, and cat scratches or bites transmit it. Usually, it is characterized by the development of a papular lesion at the site of inoculation accompanied by proximal regional lymphadenopathy, and patients usually develop a fever with non-specific symptomatology. Atypical clinical presentations of cat-scratch disease include Parinaud’s oculoglandular syndrome, encephalitis, endocarditis, and fever of unknown origin [105].

Diagnosis of bartonellosis is usually clinical. However, some laboratory methodologies can be performed to confirm the etiological diagnosis. Although *Bartonella* is a fastidious bacterium, it can be isolated by blood culture, and it can also be observed in peripheral blood smears during the acute phase of *B. bacilliformis* infection (Oroya fever). Serological methods such as ELISA and IFA can detect IgM or IgG antibodies and evidence seroconversion if acute and convalescent samples are available. The bacterial DNA can be detected by performing PCR on blood and lymph node aspirates, mainly in cases with high suspicion but negative culture attempts [106,107].

In Colombia, Carrion’s disease was unknown until 1936, when a high-mortality outbreak of febrile illness associated with severe hemolytic anemia took place among the indigenous population in the southwest of Nariño department and lasted until 1941, with more than 6000 cases [108,109]. Although similar cases have been diagnosed in other parts of the country, such as Bogotá, D.C. [110], these patients had a history of travel to the endemic area in Nariño. Late in 1941, novel cases occurred in villages on the border of the department of Cauca [111], which were close to the endemic area in the Nariño department. No confirmed cases out of the endemic area were reported until 1988, when an anemic febrile patient from Pradera municipality, Valle del Cauca department, was diagnosed with bartonellosis using microscopy [112]. Although the vector of Carrion’s disease in Colombia is unknown, historical data suggest that *Lutzomya colombiana* could be one of the vectors [112]. Despite the historical data on the presence of Carrión’s disease in Colombia, no novel cases have been reported. Several hypotheses have been raised in order to explain the origin and extinction of Carrion’s disease in Nariño. Apparently, the disease may have been imported from Peru with the entry of Peruvian soldiers into the Nariño region during the Colombia–Peru postwar period, and the extinction of the epidemic might have been due to the probable exhaustion of human reservoirs, considering the high lethality of the disease [113]. According to historical data, Carrion’s disease was prevalent in Colombia’s South Pacific subregion in the department of Nariño and surrounding areas in Cauca department. Thus, it is necessary to include Oroya fever among the differential diagnoses of malaria and other febrile illnesses, and the “verruga peruana” among the dermatological lesions in this region.

Since 2000, only a few studies have been performed to determine the exposure to *Bartonella* in Colombia. Studies performed among rural workers from several municipalities of Córdoba and Sucre departments [88], inhabitants of Monteria and Cereté municipalities (Córdoba department) [91], and homeless population from Bogotá, D.C. [92] evidenced different *Bartonella* spp. seropositivity rates (Table 2). Furthermore, the pathogen *B. quintana* was confirmed in clothing lice collected from homeless people from Bogotá, D.C. [92]. Although no confirmed cases of *B. henselae* infection have been reported to date, clinical cases compatible with cat-scratch disease in young children have been reported in Medellin, Antioquia department [114], and in Cali, Valle del Cauca department [115], as well as a Parinaud’s oculoglandular syndrome in a hospital in Medellin, Antioquia (Figure 1F) [116]. These bacteria could be circulating in domestic animals; however, these data remain unknown, as only one study has evidenced exposure to *Bartonella* spp. in 10.1% (26/258) of dogs from Bogotá, D.C., using antigens of *B. henselae*, *B. clarridgeiae*, and *Bartonella vinsonii* subsp. *berkhoffii* [117].

## 8. Relapsing Fever Group *Borrelia*

*Borrelia* is a bacterial genus of Gram-negative spirochetes. They can be classified into two groups: the *Borrelia burgdorferi* sensu lato complex, which includes at least 20 different genospecies, and the relapsing fever group, which includes many species [118]. Both of them are implicated in human infectious diseases, causing two well-described illnesses: Lyme borreliosis, caused by *Borrelia burgdorferi* sensu stricto, *Borrelia afzelii*, and *Borrelia garinii* [119], and relapsing fever, which is caused by several relapsing fever group *Borrelia* spp. [120].

Relapsing fever is an episodic AUFI that can be transmitted by the human clothing lice or ticks of the Argasidae and Ixodidae families, with the genus *Ornithodoros* spp. being the most important in Latin America [120,121]. The disease develops abruptly as a febrile illness with chills and other non-specific symptomatology, lasting for approximately three to five days, followed by an afebrile period that ends with further febrile episodes. Usually, three to thirteen recurrences can occur; the highest number of febrile recurrences occurs when the disease is left untreated; however, the severity of the febrile episodes usually decreases with every recurrence. Significant organ involvement can also occur, which is most frequent in louse-borne relapsing fever, resulting in a poor prognosis with a high mortality risk when organ damage is severe [120]. Tick-borne relapsing fever is a zoonotic disease for which many animal species might act as reservoirs. The principal risk factors for acquiring the disease are poverty and overcrowding for louse-borne relapsing fever and performing activities that increase the risk of tick exposure for tick-borne relapsing fever [120].

For the relapsing fever group *Borrelia*, diagnosis can be made by visualizing the bacteria in blood samples using dark-field microscopy or stains such as Wright–Giemsa during the febrile paroxysms; further, the detection of the bacterial DNA using amplification with PCR can be performed [122].

In the Americas, cases of relapsing fever have been reported in many countries from North, Central, and South America. The distribution of the disease and the implicated *Borrelia* species may depend on the geographical distribution of its tick vector [121,123].

The first documented case of tick-borne relapsing fever in Colombia was reported in 1906 in Villeta municipality (Cundinamarca department) [124]. During the same year, a febrile illness due to yellow fever and tick-borne relapsing fever occurred at Muzo mines in Boyacá department and lasted until early 1907, with a fatality rate of 20% [125,126]. One year later, in 1907, novel cases of tick-borne relapsing fever were reported in Manizales municipality (Caldas department) [127]. In 1923, new cases of relapsing fever were reported in Bucaramanga city (Santander department), and due to these findings, a tick collection across different regions of Colombia was made, detecting *Borrelia* spp. from *Ornithodoros rudis* collected in Buenaventura, Barranquilla, Bucaramanga, Tumaco, Quibdó, Lloró, and Itsmina [128]. In 1928, 91 cases of relapsing fever diagnosed using microscopy, of which 29 were foreigners and 62 were Colombians, were reported in San Juan (Chocó department) [129]. In 1934, novel cases were reported in Villeta and Puerto Lievano municipalities (Cundinamarca department) [124]. Subsequently, cases have increasingly been reported in other regions in Caldas, Cauca, Tolima, and Risaralda (Figure 1G) [130].

Seropositivity against *Borrelia* spp. using *B. burgdorferi* antigens has been reported in patients in a hospital in Cali (Valle Del Cauca) [94]; rural workers from the municipalities of Monteria, Cereté, Lorica, and Cotorra (Córdoba department) [93]; and febrile patients from Puerto Berrio, Puerto Nare, and Cimitarra municipalities (Antioquia and Santander departments) [90]. However, these serological data must be interpreted carefully due to cross-reactivity among *Borrelia* spp. [122,131]. Thus, it is likely that these results were due to exposure to other *Borrelia* species of the relapsing fever group or other infectious agents, such as *Treponema* species.

Attempts to detect *Borrelia* spp. in animals have been made, finding positivity using molecular methods in bats from Macaregua cave, Santander department [132,133], and bats captured in different municipalities (Montelíbano, Tierralta, San Antero, Montería, Lorica, and Moñitos) in Córdoba department [134] forming a new taxon. Furthermore, in Córdoba department, the presence of the anthropophilic tick *Ornithodoros puertoricensis* was described [135], in which in Panama, a novel species of the relapsing fever group borreliae (*Borrelia puertoricensis*), of unknown pathogenicity, was detected [136]. All these findings highlight the need for more research to understand *Borrelia*’s local epidemiology. However, there are enough historical data to state that relapsing fever must be included in the differential diagnosis of AUFIs in Colombia.

## 9. *Coxiella burnetii*

*Coxiella* is a bacterial genus composed of intracellular Gram-negative rods [137]. To date, only *Coxiella burnetii* has been associated with human disease. It has two antigenic phase variants: the phase I variant, which is the virulent form of the bacteria that can be found in nature and predominates in chronic forms of the disease, and the phase II variant, which is an attenuated form of the bacteria and is associated with acute forms of disease [137,138]. *C. burnetii* is an aerosol-borne pathogen; its natural form is highly resistant to chemical agents and extreme environmental conditions. Thus, it is considered extraordinarily infectious and one of the most feared potential bioterrorism agents [139].

The disease caused by *C. burnetii* is called Q fever (query fever). It mainly affects people living in rural areas in contact with cattle [138]. Q fever has an incubation period of two to three weeks, and it does not have a typical form, being clinically variable between patients and making its diagnosis challenging [140]. Q fever is a self-limited acute febrile illness accompanied by headaches, myalgias, arthralgias, and cough as the principal non-specific clinical manifestations [138,141]. Although the disease can be mild in most cases, some patients develop severe and chronic forms of the disease, which include prolonged fever, neurologic signs, hepatitis, atypical pneumonia, and myocarditis associated with pericarditis, with the latter two being the leading causes of death [138,141].

For laboratory confirmation of Q fever, paired serum samples must demonstrate seroconversion with IgG IFA using *C. burnetii* phase I and II antigens. IgM antibodies can also be requested but are less specific; thus, they only provide limited diagnostic values without IgG results. PCR can also test blood or serum samples during the disease’s acute phase to detect the DNA of *C. burnetii*. Isolation of the pathogen can be performed in cell cultures; however, it is challenging and requires a Biosafety Level 3 laboratory, since its manipulation can be hazardous and laboratory transmission has already been reported [142]. Chronic Q fever can be confirmed with the detection of anti-phase I IgG antibodies ≥ 1:1024; tissue biopsies can also be tested using PCR or IHC [142].

In Colombia, four cases of Q fever have been reported to date. The first one occurred in 2012 and developed as culture-negative infectious endocarditis in a farmer from the municipality of Nariño, Antioquia department; the disease was accompanied by splenic infarction, retinal emboli, and renal failure; the diagnosis of Q fever was made using serology, finding positive titers of phase I antibodies [143]. Another case, which developed as acute atypical pneumonia in a patient who may have acquired the infection in a rural area in the municipality of Manizales, Caldas department, was reported in the same year. The diagnosis was made with serology, ruling out other atypical microorganisms [144]. A third case was reported in 2014 in a patient with a background of livestock contact while working as a farmer in a tropical rural area of Monteria municipality, Córdoba department. The patient was enrolled during a surveillance epidemiological study and was clinically asymptomatic but with mild signs and symptoms of heart damage. The diagnosis was made with serology, which evidenced elevated titers of phase I and phase II antibodies [145]. Finally, in 2021, a case of acute Q fever with generalized maculopapular and purpuric rash was reported by a zootechnician who may have acquired the infection while he was working with goats that had recently given birth in a tropical rural area in the Magdalena Medio region, Antioquia department; it was diagnosed using PCR [146].

Several seroprevalence studies have been performed in different regions of Colombia, evidencing different seropositivity rates among slaughterhouse personnel from the city of Medellin [96] and livestock farmers from the North and Magdalena Medio regions [95] and San Pedro de Los Milagros municipality [89] in Antioquia department. Seropositivity was also found among livestock farmers from rural areas of Monteria (Córdoba department) [98] and inhabitants of San Marcos, Cotorra, Lorica, Monteria, and Cienaga de Oro municipalities (Córdoba and Sucre departments) [88]. Additionally, molecular detection of *C. burnetii* was evidenced among asymptomatic farmers from Puerto Berrío, Puerto Nare, and Puerto Triunfo (Antioquia department) (Table 2) (Figure 1H) [97]. These data suggest that *C. burnetii* is actively circulating in Colombia.

Contact with domestic livestock animals, including cattle, goats, and sheep, may represent the main risk for acquiring the infection, as evidence of *C. burnetii* has been detected in these animals from Colombia [95,147] and their products [98]. Furthermore, *C. burnetii* may also be circulating in a wild cycle, as evidence of the bacteria was found among bats that inhabit Macaregua cave, Santander department, Colombia [148]; thus, cavers and people who perform eco-touristic activities may be aware of the risk that Q fever may represent for them. Q fever remains a minor suspected etiology of febrile illness in Colombia; its symptomatology is not characteristic; and diagnostic methods for its confirmation are not offered routinely, which is probably why few reported cases and high human seropositivity rates exist.

## 10. *Ehrlichia*

*Ehrlichia* genus is composed of obligate intracellular Gram-negative pleomorphic tick-borne bacteria, which infect a wide range of host cells, including endothelial and hemopoietic cells (e.g., macrophages and neutrophils), in which they form clusters known as morulae [149]. To date, *Ehrlichia chaffeensis*, *Ehrlichia ewingii*, and *Ehrlichia muris* subsp. *eauclairensis* have been implicated as human pathogens and are restricted to specific areas of the United States due to the distribution of their tick vectors. Nevertheless, novel species are continuously being detected from several animal hosts. It is likely that some of these may have a zoonotic potential [149,150,151]. On the other hand, several authors have suggested that *Ehrlichia canis*, a pathogen of dogs, may also be linked to human disease [152]; however, to date, this is still controversial, and more studies are needed to clarify it.

*E. chaffeensis* is the primary human pathogen; it causes human monocytic ehrlichiosis (HME), which is a mild non-specific febrile illness [153]. However, in 15% of patients, a severe form of the disease can be developed with meningoencephalitis, acute renal failure, myocarditis, lung damage, gastrointestinal hemorrhage, and coagulopathies [153,154]. Little is known about the disease caused by the other two pathogenic species, but reports from case series suggest that they may cause a milder disease similar to HME [155,156].

Laboratory diagnosis of ehrlichiosis is similar to anaplasmosis and should be performed in clinically compatible cases with high-risk factors. The reference method is IFA, with which IgG seroconversion must be evidenced. Amplification with PCR of the bacterial DNA extracted from blood samples can also be used for diagnosis in the early stages of the disease; culture isolation is also possible, but it is only performed in specialized laboratories; and direct visualization of morulae in peripheral blood samples using microscopy can be highly suggestive mainly in risk populations [79,157].

In Colombia, three presumed cases of human ehrlichiosis have been reported to date, but two lack reliable diagnosis methods to be classified as confirmed cases [158,159]. The only convincing reported case is a febrile pediatric patient from a rural area of central-western Colombia who had come into contact with dog ticks and was diagnosed using qPCR [160].

Furthermore, several serological studies have evidenced exposure to *Ehrlichia* spp. in Antioquia department among livestock farmers from the North and Magdalena Medio regions [95] and San Pedro de Los Milagros municipality [89], and among febrile patients from the Magdalena Medio region (Table 2) (Figure 1I) [90]. In addition, *E. canis* has been confirmed from dogs samples in several regions, including the Cauca department [161], where a comprehensive study was performed, revealing that humans are not usually infected by this species [162]. Thus, these findings should reinforce surveillance studies to understand if *Ehrlichia* spp. could be an emerging human pathogen in Colombia.

## 11. *Orientia*

*Orientia* spp. are a bacterial genus formed by small intracellular Gram-negative rods transmitted by chiggers, the larval stage of trombiculid mites [163]. Classically, *Orientia tsutsugamushi* was the only recognized species for a long time; however, recent studies have proposed two novel species, *Candidatus* Orientia chuto and *Candidatus* Orientia chiloensis, both pathogenic for humans [164].

The disease caused by these species is known as scrub typhus. This re-emerging vector-borne infectious disease develops as an AUFI associated with rash and a necrotic eschar at the site of inoculation [165]. The disease can be mild or severe; in the latter, respiratory, neurological, renal, gastrointestinal, and cardiovascular damage can occur, compromising the patient’s life [166].

For confirmatory diagnosis of scrub typhus, a four-fold rise in seroconversion of IgG antibodies between acute and convalescent serum samples using IFA is required. However, ELISA detection of IgG or IgM antibodies can also be performed [167]. Molecular methods such as PCR can also be performed to detect bacterial DNA. In some cases, bacterial DNA can be detected in whole-blood samples. Using an eschar sample or a swab of the uncrusted lesion is preferable, as it is more likely to detect bacterial DNA due to the high load of bacteria in them [168]. Isolation can also be performed; however, it can be only performed in reference laboratories [167].

Classically, scrub typhus was restricted to the Asia-Pacific region in a geographic zone known as the “tsutsugamushi triangle,” which has been known as the only endemic region worldwide [169]. However, several studies have evidenced autochthonous cases from the United Arab Emirates and exposition to *Orientia* spp. in several countries of Africa [170,171]. In Latin America, a newly endemic region has been found in Chile, where seropositivity and autochthonous scrub typhus cases have been evidenced, as well as the vector involved in transmitting the disease [9,172,173,174]. Since then, serological evidence has been found in Peru and Honduras, suggesting that scrub typhus might be more widespread than initially thought and could be one of the etiologies of AUFIs in Latin America [164,175,176].

In Colombia, no attempt to detect the bacterium or the exposition to it was made until 2022, when a study found seropositivity to *Orientia* spp., confirmed with Western blot, among inhabitants of four municipalities of Cauca department (Table 2) (Figure 1J) [99], which should encourage to perform more studies to establish if *Orientia* is part of febrile illnesses in the region, and if so, determine which species are involved, as well as its possible implications for Colombian public health or whether further surveillance by local health systems is warranted.

## 12. *Rickettsia*

*Rickettsia* spp. is a bacterial genus of the order Rickettsiales and family Rickettsiaceae, which includes small obligate intracellular Gram-negative short rods [177]. These can be classified into four groups: the spotted fever group (SFG), the typhus group (TG), the transitional group, and the ancestral group, of which the first three include several human pathogenic species [178]. A total of 27 *Rickettsia* spp. have been described, and at least seventeen of them are recognized as human pathogenic species [179], of which, to date, *Rickettsia rickettsii*, *Rickettsia typhi*, *Rickettsia prowazekii*, and *Rickettsia akari* are the most important. However, emerging species, such as *Rickettsia parkeri*, and geographically restricted species, such as *Rickettsia conorii* and *Rickettsia africae*, are also highly important pathogenic species [180].

Clinically, rickettsioses are among the most important and underrated causes of AUFIs in several regions worldwide [181]. SFG rickettsioses, transmitted by ticks, develop as febrile illnesses accompanied by non-specific symptoms with the development of rash, usually macular or maculopapular, which is highly variable and depends on the pathogen. Some species can induce the formation of an eschar at the site of inoculation, which can be single or multiple; fatality is variable and depends on the species, with *R. rickettsii* being the deadliest one and generating, in severe cases, pulmonary, renal, and neurologic damage [180,182]. TG rickettsioses, caused by *R. prowazekii* (epidemic typhus transmitted by lice) and *R. typhi* (murine typhus transmitted by fleas), are also febrile illnesses with non-specific symptoms, among which rash and gastrointestinal symptomatology such as nausea and vomiting can be present in about half of the cases. Murine typhus is generally mild, but epidemic typhus can be severe, with a case fatality rate as high as 50% due to complications that include endocarditis, pneumonia, renal failure, meningitis, and septic shock, among others [180,183,184].

Laboratory diagnosis of rickettsioses is complex and only sometimes available in endemic areas. The gold standard method for confirmatory diagnosis is IFA for IgG antibodies, which need to be performed on paired serum samples to evidence a four-fold seroconversion. Only single-sample results cannot be used for laboratory confirmation, as antibody titers can be observed in persons even four years after the acute illness, mainly in endemic areas. However, a single titer of at least 1:256 obtained during a clinically epidemiologically compatible illness can also suggest rickettsiosis [185,186]. PCR amplification can also be a helpful tool. Detection of bacterial DNA in whole-blood samples cannot always be performed, as *Rickettsia* spp. infect endothelial cells and do not circulate in large numbers in the bloodstream until the disease has progressed severely. However, samples such as a skin biopsy of a rash lesion, the inoculation eschar, or a swab sample of the uncrusted lesion can be used for PCR amplification, as these samples have a high bacterial load [186,187]. *Rickettsia* can be isolated; however, isolation can only be performed in specialized laboratories [186].

In Colombia, for the period from 1902 to 1986, there are many historical data regarding the presence of different rickettsioses, which include cases of SFG, TG, and unspecific rickettsioses in several regions of the departments of Antioquia, Atlántico, Bolivar, Boyacá, Caldas, Cauca, Cundinamarca, Córdoba, Huila, Magdalena, Meta, Nariño, Norte de Santander, Santander, Tolima, and Valle Del Cauca [188]. Since then, a significant number of studies have been performed in several regions in the last twenty years, establishing at least four endemic regions for rickettsioses: Tobia valley in Cundinamarca department, the north region of Caldas department, the Urabá region in Antioquia department, and the Caribbean region in Córdoba department [189]; however, other regions, such as the Orinoquia region, may also be considered crucial, as it is probably endemic [190,191].

In Cundinamarca department, the first cases of SFG rickettsiosis due to *R. rickettsii* were described in 1935 by Dr. Luis Patiño Camargo, when an epidemic of the disease occurred in the town of Tobia, in which 65 cases were reported, of which only 3 recovered, giving a high fatality rate of 95% [192]. Years later, in 1941, two novel fatal cases from areas close to village of Tobia were reported [193]. Since then, an epidemiological silence occurred until 2007, when two novel fatal cases of *R. rickettsii* rickettsiosis were reported in Villeta municipality, near the same region where the disease was initially discovered [194]. Several years later, fifteen non-fatal cases of SFG rickettsioses were detected in the same municipality during a febrile illness surveillance study, of which twelve had co-infections with dengue or leptospirosis [85]. Due to the high fatality rate of *R. rickettsii* rickettsiosis, febrile cases detected in Tobia valley and adjacent areas should be carefully managed, as this region is highly endemic for this pathogenic species.

In the department of Caldas, the first reports of confirmed cases were performed in 1942, 1943, and 1946, all reported as TG rickettsioses [188]. Since then, novel confirmed cases have been reported in 2008, when a study detected fourteen murine typhus cases in the department’s north part [195]. Several years later, a prospective study performed in the same region, specifically in Aguadas, Filadelfia, and Salamina municipalities, evidenced several SFG and TG rickettsioses [196]. Finally, in another surveillance study, eight febrile patients enrolled in the municipalities of Belalcázar, Salamina, Filadelfia, Noracasia, and San José were confirmed as SFG and TG rickettsioses [197]. These studies confirm that murine typhus and at least one pathogenic SFG *Rickettsia* sp. are circulating in Caldas department.

In the Urabá region, department of Antioquia, the first reports of rickettsioses cases occurred in 2006, when an outbreak of *R. rickettsii* infection was reported in the municipality of Necoclí, which had a fatality rate of 35% [10]. In addition, during a non-malarial febrile illness surveillance study, nine cases of rickettsioses, where three of them were co-infections with other febrile illness etiologies, were detected among inhabitants of three municipalities: Apartado, Necoclí, and Turbo [21]. Years later, another study also found six confirmed cases of SFG rickettsioses among febrile patients from the same region [198]. The first case of rickettsiosis due to the emerging pathogen *R. parkeri* strain Atlantic Rainforest was also confirmed in the municipality of Turbo [199]. Fatal cases have also been reported in the region, with the first one being a fatal pediatric case due to *R. rickettsii* in the municipality of Chigorodó [200], and the second one being a fatal case due to co-infection between *R. rickettsii* and *Leptospira interrogans* serovar Copenhageni in a patient from Carepa municipality [201]. These studies clearly show that the Urabá region is an endemic region for at least two different SFG rickettsioses. Furthermore, not so far away from the Urabá Antioquia region, in the municipality of Uramita, four cases of *R. rickettsii* infection confirmed with laboratory and epidemiological association have also been reported, suggesting that it may also be considered an important region in which rickettsioses may be circulating [202,203].

Finally, in the department of Córdoba, the first report of an unspecific rickettsiosis took place in 1947 [188]. Since then, novel cases occurred in 2011, when twenty cases of *R. rickettsii* infection were detected in the municipality of Los Córdobas; four of these died, giving a fatality rate of 20% [204]. Years later, two SFG rickettsioses cases were identified in Monteria municipality during an AUFI surveillance study [48]. Although only two studies have evidence of the presence of cases of rickettsioses in this region, the fact of having detected cases due to *R. rickettsii* and evidence of seropositivity to SFG *Rickettsia* spp. in several municipalities of the north of the department [205] should alert local authorities to consider this region as highly endemic for this pathogen.

Besides these regions, several cases of rickettsioses have been detected in other regions. In the department of Quindío, fifteen cases of rickettsioses, with five of them being co-infections with dengue, were detected among febrile patients in several health institutions [206]. In the department of Meta, a probable case of SFG rickettsiosis with seroconversion was reported in a patient who had recently traveled to the municipality of Puerto Gaitán [190]. Finally, in the department of Valle Del Cauca, a probable case of murine typhus was reported in a febrile patient from an urban area of the city of Cali (Figure 1K) [207]. All these data reinforce the need for a better monitoring of rickettsioses in the country and further studies to establish if other regions are endemic.

## 13. Conclusions

Although the Colombian healthcare surveillance system monitors a significant number of zoonotic infectious diseases, several are still underrated and neglected, and some even remain undiscovered. With this review, which collected information until early 2023, we found evidence that some etiologies of non-notifiable zoonotic tropical febrile illnesses are present in several regions of Colombia, generating a risk for the local population, travelers who visit the country, and immigrants. Due to the close relationship among humans, domestic animals, and wildlife in several scenarios related to habitat destruction (e.g., intensive agriculture, accelerate urbanization, and eco-tourism) and climate change, zoonotic febrile illnesses represent a significant public health problem. These diseases negatively affect the population’s well-being, since their clinical suspicion and confirmatory diagnosis are not always available. Thus, their actual contribution to the disease burden remains unknown. It is thus necessary to carry out more studies using the One Health approach, to understand the importance of these diseases for local eco-epidemiology and public health. Different research groups, with different national and international funding, contributed to the knowledge of these infectious diseases on the national territory. However, despite this, the local surveillance system remains unaware of the importance of these illnesses for local public health, making it necessary to keep promoting and suggesting their inclusion into the mandatory notification system.

## Figures and Tables

**Figure 1 microorganisms-11-02154-f001:**
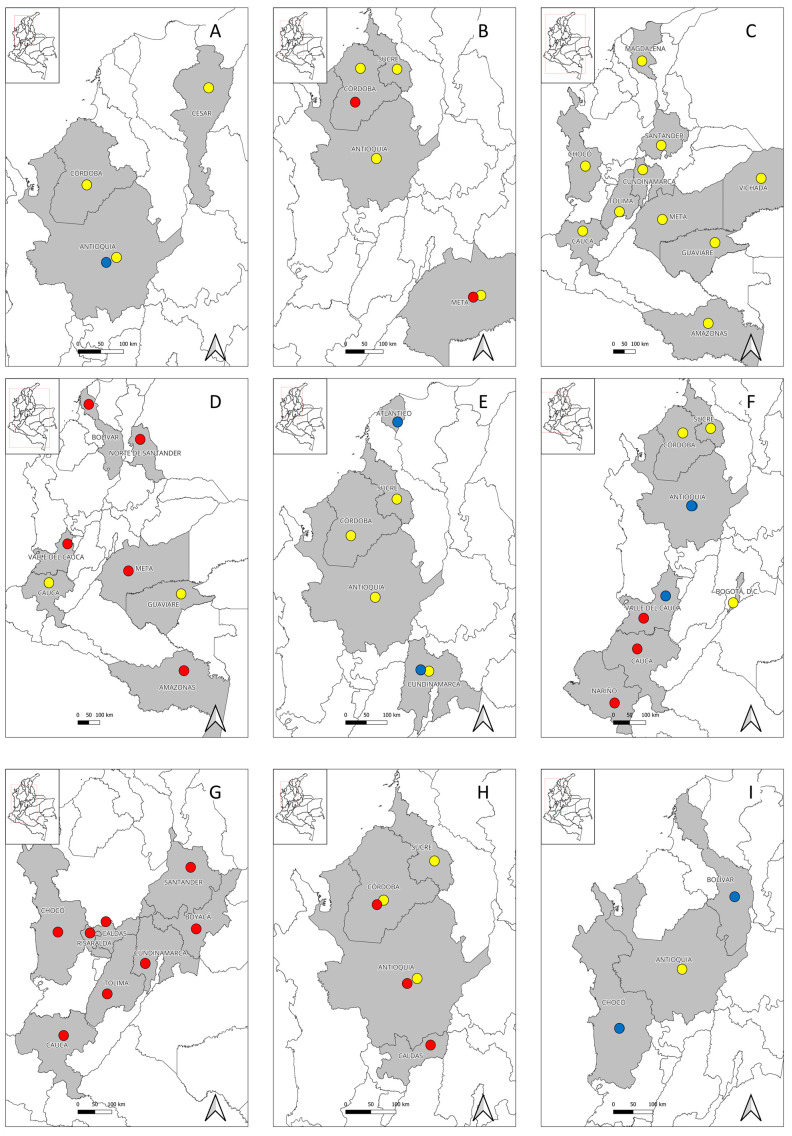
Geographic distribution of human cases and regions with serological or molecular evidence of etiologies of zoonotic tropical febrile illnesses that are not part of the notifiable diseases in Colombia as of early 2023. (**A**) Arenavirus, (**B**) Hantavirus, (**C**) Mayaro virus, (**D**) Oropouche virus, (**E**) *Anaplasma*, (**F**) *Bartonella*, (**G**) relapsing fever group *Borrelia*, (**H**) *Coxiella burnetii*, (**I**) *Ehrlichia*, (**J**) *Orientia*, and (**K**) *Rickettsia*. Research studies were mainly carried out in regions of the northwest of the country. The presence of these infectious agents in other regions remains unknown. Departments marked in gray indicate evidence of human cases or regions with serological or molecular evidence of the microorganism. Yellow dots mean regions with serological evidence. Blue dots mean reports of probable cases. Red dots mean reports of confirmed cases.

**Table 1 microorganisms-11-02154-t001:** Serological and molecular studies on Arenavirus, Hantavirus, Mayaro virus, and Oropouche virus in Colombia.

Infectious Agent	Department	Region	Population of Study	Serological/Molecular Method	Positivity (%)	Ref.
Arenavirus	Antioquia	Urabá antioqueño	Febrile patients	IgG enzyme-linked immunoassay (ELISA)	1/220 (0.5)	[21]
	Cesar	Kankuamos community	Indigenous population	IgG ELISA	2/506 (0.4)	[23]
	Córdoba	Embera Katio community	Indigenous population	IgG ELISA	10/325 (3.1)	[22]
Hantavirus	Antioquia	Urabá antioqueño	Febrile patients	IgG ELISA	2/220 (0.9)	[21]
	Cesar and Córdoba	Kankuamos and Tuchín communities	Indigenous population	IgG ELISA	5/506 (1)	[23]
	Córdoba	Chimá, Cienaga de Oro, Cotorra, Lorica, Purisima, Sahagún	Inhabitants	IgG ELISA	24/286 (8.4)	[24]
		Embera Katio population	Indigenous population	IgG ELISA	5/324 (1.5)	[22]
		Tuchín community	Indigenous population	IgG ELISA	7/87 (8)	[25]
	Córdoba and Sucre	Twelve towns	Rural male workers	IgG ELISA	12/88 (13.6)	[26]
	Meta	Not specified	Febrile patients	IgG ELISA	7/100 (7)	[27]
Mayaro virus	Amazonas	Araracuara	Indigenous population	Hemagglutination Inhibition (HAI) (cutoff 1:20)	77/396 (19.4)	[28]
	Cauca	El Tambo, La Sierra, Santander de Quilichao	Rural population	Plaque reduction neutralization testing (PRNT) (cutoff 1:20)	5/505 (1)	[29]
	Chocó, Cundinamarca, Meta, Santander, Tolima, Vichada	Not specified	Rural population	Neutralization assay, 50 Median Lethal Dose (LD50)	57/408 (14)	[30]
	Guaviare and Magdalena	Not specified	Febrile patients	HAI (cutoff 1:10)	10/54 (18.5)	[31]
	Not specified	Not specified	Military recruits	HAI (cutoff 1:20)	7/292 (2.4)	[32]
Oropouche virus	Amazonas	Leticia	Febrile patients	qRT-PCR	43/153 (28.1)	[33]
	Cauca	El Tambo, La Sierra, Santander de Quilichao	Rural population	PRNT (cutoff 1:20)	10/505 (2)	[29]
	Guaviare	Not specified	Febrile patients	HAI (cutoff 1:10)	4/54 (7.4)	[31]
	Meta	Villavicencio	Febrile patients	qRT-PCR	38/566 (6.7)	[33]
	Norte De Santander	Cúcuta	Febrile patients	qRT-PCR	3/19 (15.8)	[33]
	Valle Del Cauca	Cali	Febrile patients	qRT-PCR	3/53 (5.7)	[33]

**Table 2 microorganisms-11-02154-t002:** Serological and molecular studies on *Anaplasma*, *Bartonella*, *Borrelia*, *Coxiella burnetii*, *Ehrlichia*, and *Orientia* in Colombia.

Infectious Agent	Department	Region	Population of Study	Serological/Molecular Method	Positivity (%)	Reference
*Anaplasma*	Antioquia	San Pedro de Los Milagros	Livestock farming workers	IgG indirect fluorescent antibody test (IFA) (cutoff value 1:64)	19/328 (5.8)	[89]
	Córdoba and Sucre	Cienaga de Oro, Cotorra, Lorica, Montería, San Marcos	Rural workers	IgG IFA (cutoff value 1:64)	15/75 (20)	[88]
	Cundinamarca	Villeta	Febrile patients	IgG IFA (cutoff value 1:64)	7/104 (6.7)	[85]
	Not specified	Rural Magdalena Medio	Febrile patients	IgG IFA (cutoff value 1:64)	39/271 (14.4)	[90]
*Bartonella*	Córdoba	Montería, Cereté	Human population	IgG IFA (cutoff value 1:64)	39/80 (48.7)	[91]
	Córdoba and Sucre	Cienaga de Oro, Cotorra, Lorica, Montería, San Marcos	Rural workers	IgG IFA (cutoff value 1:64)	30/77 (39)	[88]
	Cundinamarca	Bogotá, D.C.	Homeless people	IgG IFA (cutoff value 1:64)	49/153 (32)	[92]
*Borrelia*	Antioquia and Santander	Cimitarra, Puerto Berrio, Puerto Nare	Febrile patients	IgG enzyme-linked immunoassay (ELISA)	18/271 (6.6)	[90]
	Córdoba	Cereté, Cotorra, Lorica, Monteria	Rural workers	IgG ELISA	30/152 (20)	[93]
	Valle Del Cauca	Cali	Hospital patients	Immunoblot	Not specified	[94]
*Coxiella burnetii*	Antioquia	Magdalena Medio	Livestock farming workers	IgM/IgG IFA (cutoff value 1:16)	17/143 (11.9)	[95]
		Magdalena Norte	Livestock farming workers	IgM/IgG IFA (cutoff value 1:16)	89/189 (47.1)	[95]
		Medellin	Slaughterhouse personnel	Complement Fixation Test (cutoff value 1:10)	83/153 (54) 37/153 (24) (recent infection 1:20)	[96]
		Puerto Berrio, Puerto Nare, Puerto Triunfo	Healthy farmers	qPCR	37/143 (25.9)	[97]
		San Pedro de Los Milagros	Livestock farming workers	IgG IFA (cutoff value 1:16)	20/328 (6.1)	[89]
	Córdoba	Monteria (rural areas)	Livestock farming workers	IgG IFA (cutoff value 1:64)	37/61 (61)	[98]
	Córdoba and Sucre	Cienaga de Oro, Cotorra, Lorica, Montería, San Marcos	Rural workers	IgG IFA (cutoff value 1:16)	17/72 (23.6)	[88]
*Ehrlichia*	Antioquia	Magdalena Medio	Febrile patients	IgG IFA (cutoff value 1:64)	73/271 (26.9)	[90]
			Livestock farming workers	IgM/IgG IFA (cutoff value 1:16)	51/143 (35.7)	[95]
		Magdalena Norte	Livestock farming workers	IgM/IgG IFA (cutoff value 1:16)	148/189 (78.3)	[95]
		San Pedro de Los Milagros	Livestock farming workers	IgG IFA (cutoff value 1:64)	42/328 (12.8)	[89]
*Orientia*	Cauca	Caloto, Santander de Quilichao	Inhabitants	IgG ELISA IgG IFA (cutoff value 1:128)	67/486 (13.8)	[99]

## Data Availability

Not applicable.

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
