# Peer review of "Etiologies of Zoonotic Tropical Febrile Illnesses That Are Not Part of the Notifiable Diseases in Colombia"

_microorganisms, 2023, doi:10.3390/microorganisms11092154_

Round 1

Reviewer 1 Report

Comments:

1. Line 242, 426: The abbreviation” AUFI” should be showed full name initially.

2. Line 247:  “C. paraensis and C. quinquefasciatus “,  The C genus is different for each, so C. quinquefasciatus might be as Cx. quinquefasciatus for differentiation.

3. Line 484, 485: In ref 126, only find a novel Ornithodoros tick species, no Borrelia was detected.

4. Line 682: Of the [178,180,181], the ref [181] was not related.

5. Line 723: The abbreviation” RMSF” should be showed full name initially.

6. Line 791-796: Fig. 1. The A~K alphabet represent each etiology of tropical fever diseases in Colombia, it might clear in this figure if each eleven etiologies was printed with each alphabet.

1. Line 472: “this cases were expositions to other Borrelia species ….”  Need Minor editing

2.Line 511: “performed on paired serum samples to evidence sero-conversion” Need Minor editing.

Reviewer 2 Report

Peer Review of:

Zoonotic tropical febrile illnesses etiologies that are not part of the compulsory-reporting diseases in Colombia

Silva-Ramos et al.

This review by Silva-Ramos et al, outlines 11 diseases which are not part of the compulsory reporting system in Colombia but may be a cause of febrile illnesses and are possibly under- reported or not always recognised. While it comprehensively describes these diseases, I find at times there is little synthesis of the data or discussion of the implications of these findings for public health practitioners or clinicians within Colombia. It is not always clear why these 11 diseases (Anaplasma, Arenavirus, Bartonella, Borrelia, Coxiella burnetii, Ehrlichia, Hantavirus, Mayaro virus, Orientia, Oropouche virus and Rickettsia) were chosen and the focus on some diseases with only possible or speculative cases but little current public health implications seem misplaced. Sometimes a paragraph or two would suffice rather than an in-depth focus. Some findings of extensive seropositivity rates to diseases that have only a few case reports are somewhat alarming and suggests under recognition of some diseases which have significant implications. Rather than in depth discussion of each disease- I think a more narrative approach of possible public health threats in Colombia (which are not currently on the compulsory reporting list) and the implication of these would be more useful. There needs to be discussion of how the country should respond to these threats and the challenges of implementing further surveillance systems and education campaigns for example. Why should public health authorities and medical practitioners be concerned about these diseases is not always clearly elucidated. It seems clear (I think mostly rightly so) that the authors are concerned about these zoonotic diseases but there is very limited discussion on how they propose the country could potentially prevent these diseases from becoming major public health issues. Apart perhaps from a change in focus, there are extensive language and grammar issues, which need attention, I think this review could be reduced in size while still maintaining much useful information for readers in both Colombia and elsewhere. A revised manuscript would be an informative and useful article but as it stands it is hard to determine the overall aims of the authors rather than just describing these 11 diseases, which is not overly useful.

General comments:

Suggest revise title – it is long and cumbersome!

Throughout the manuscript there needs to be review of grammar / English. Consider an independent review by a native English speaker.

Many of these pathogens may be unfamiliar to readers or testing for them may be rarely requested – it might be worth putting in a comment of the diagnostic capabilities of the country – is there a reference lab for these diseases – are there commercial assays available for various diseases mentioned or is there a need to develop in house assays? Do any of the tests need to be sent to another country – such as testing at the CDC. Are there any financial implications of this.

For Mayaro virus- it is stated “several regions are in greater risk of a possible MAYV epidemic once the virus adapts to an urban cycle, thus, MAYV might be one of the most important threats in the future”. This could be discussed more, given the seroprevalence studies, is there much asymptomatic disease? What could lead to an urban cycle and has this occurred in other countries?

Oropouche virus, while known to be in primates for decades, the first human case was diagnosed in 2021- with seroprevalence of 2% in one study and PCR detection up to ~31% in acute febrile illness from 2019-2022! This finding by KA Ciuoderis et al is alarming and I think deserves further discussion. Have there been any specific environmental changes leading to the emergence of this disease, or has it just not been recognized for some time.   

Anaplasma / Bartonella /Ehrlichia– Consider reducing some background information on these diseases, I think these are less likely to constitute a major public health threat to Colombia for various reasons, but clinicians should maybe be aware of them as potential pathogens in the country. The inclusion of Ehrlichia and discussion in such detail is a bit unusual as the cases discussed do not seem very convincing.

Reduce discussion on Lyme disease.

For Q fever- it is interesting to note the high rates of sero-prevalence but generally low reported cases in humans. The reasons for this are not really discussed in the manuscript.

The conclusion and discussion of the implications of the seroprevalence studies is very limited and I think much more focus should be on the implications of finding relatively unknown diseases that seem to be circulating in Colombia. While maybe not causing a high burden of disease currently – it needs to be discussed what can be done about this. How could future surveillance studies/ systems be put in place? How will this be funded? Are education / public health discussions needed to inform Drs. and the public about these potential health threats. Are habitat destruction/ changing climate potential impacting factors leading to more zoonotic diseases? Would taking a One-Health approach be worthwhile? How are the diagnostic capabilities of the country set up to test for these various diseases?

At current it is hard to determine the overall impression (without going through the whole text) of the extent of circulation of the various diseases in human populations in Colombia- It might be useful to summarize all the seroprevalence studies for the different diseases within a table which briefly outlines the studies and the rates/ range of seroprevalence obtained and any significant other factors such as geographic locale, significant limitations etc. and list the study references that pertain to these. You could also summarize the known cases, but I think this would be slightly less useful.

Specific comments:

Line 32-33: “Colombia is a Latin American country considered as the second most diverse country on the planet with several different climates [1]”- Diverse in what way? The comment of several different climates contradicts the description in the abstract. While near the equator – Colombia has a range of geographic features such as coastal regions and mountainous areas at altitude – is this what you mean?

Line 35-38: This is a long and confusing sentence – please revise. Please clarify what you mean by isothermic if you are to use this term.

Line 38-40: “The most part of the country, at least 85% of the national territory, are composed of tropical and sub-tropical areas with hot and humid climates” Contradicts some of the prior statements and is poorly written.

Line 42-45: Poorly written with repetition.

Line 76: “associated with”

Line 77-80: Revise grammar and English

Line 81-87: Define IFA abbreviation. I do not think you necessarily need to point out it is a real time, reverse transcription PCR here- consider just rather just state the virus can be detected by PCR. For the purposes of this review, specific technical aspects of the tests have less relevance. Mention that virus isolation is rarely performed due to the biosafety risk of this including the need for BSL4 laboratories and the fact it is resource intensive when PCR is usually much easier to perform. You mention semen as a potential sample that can be tested which seems a bit odd- is this actually done in any circumstances?  

Line 88-94: Confusingly written, suggest rewrite. Use the term “reservoir”/“host species” rather than hosted- use the new species name and then the old one in brackets rather than the other way around. Make sure it is apparent we are talking specifically about Pichinde virus rather than arenaviruses in general.

Line 98-110: Interesting- but again this could be better written.

Line 114: as per above – “hosted” comes across as a guest in someone house for example – perhaps use “host species being rodents” or the virus reservoir is in various rodent species.

Line 115: Use commas rather than full-stops (periods)- 150.000 to 200.000 people worldwide annually

Line 124-135: This could be better written.

Line 139-140: “viral DNA from blood or tissue samples can be evidenced by RT-PCR and so, its detection can be considered a confirmatory diagnosis of hantavirus infection”- don’t you mean RNA? I would say PCR can be used in the diagnosis as a direct detection method.

Line 140-142: IHC of course requires a biopsy – is this practical? What would you perform a biopsy on?

Line 159: You do not really seroconvert to IgM as such given it is acute phase and not too specific with much cross reaction.

Line 200: “Period of free hyperthermia”- do you mean an afebrile period?

Line 217: “Psorophora mosquitoes collected in the municipality of San Vicente de Chucurí, Santander department”- this is a different genus to that mentioned above- what is the significance of this? Do these species bite humans or only animals?

Line 237- “MAYV might be one of the most important threats in the future”. This requires more discussion, clarification. Is this due to adapting to urban mosquitos, deforestation, encroachment of previously forested areas? Is there a high level of asymptomatic / pauci symptomatic disease given the relatively high sero-positivity studies (although correlation to illness might be hard to make in these studies).

Line 264- Abbreviate HAI not HI

Line 267- This is an RNA virus?

Line 318-319: So, if the vector is not present in Colombia how was this patient supposed to have acquired the disease? This could do with more explanation.

Line 302-308: I’m not sure if it really is worth expanding the information on A capra if it does not seem to present in Colombia or an immediate threat.

Line 320: Exposition? Do you mean exposure? And if so, how? Previously it was stated the required tick vectors are not present.

Line 348: Courses with acute anemia> causes acute anaemia?

Line 353: ? auto-limited? What do you mean by this?

Line 365-369: This needs to be rewritten please.

Line 370-371: “High mortality anemic febrile illness and bloody warts”- this is unusual working, please revise.

Line 379-381: “No studies tried to determine the vector of Carrion’s disease in Colombia, but likely Lutzomya columbiana, which is also an anthropophilic species as Lutzomya verrucarrum, may be implicated as vector of B. bacilliformis in endemic areas of Colombia” – Why? This is also badly written.

Line 386: “During the postwar period”- which war?

Line 388-389: “However, the possibility that Carrion's disease could have a broader distribution in Colombia must not be ruled out”- why not? Your description does not make this out to be a significant public health issue.

Line 390-391: “South Pacific region of the country”- Not sure what you mean by this as Colombia is nowhere near the South Pacific. Maybe you mean the coastal area next to the Pacific Ocean.

Line 400: “Body louse pools” not sure what this means?

Line 417-425: I do not think you need to describe Lyme disease in detail for this paper has it has little relevance to febrile illnesses acquired in Colombia.

Line 427: Argasidae and Ixodidae families- I would just keep this consistent and say Ixodes spp. Maybe refer to a specific genus for the Argasidae ticks.

Line 439-441: Why are we talking about Lyme disease? Is this a potential problem in Colombia?

Line 466-470: This sentence needs to be rewritten but I also do not think it adds much to the review as discussed this is a surprising result which needs to be interpreted with caution due to the issues mentioned from line 470-474. I would consider just leaving it out, as it most likely is just a cross reaction and of little clinical significance. 

Line 474-478: As per above comments, I really do not think this discussion on Lyme disease is relevant for the paper.

Line 491: “This genus includes three recognized species and several endosymbionts of ticks and aquatic invertebrates”- just delete not useful for purpose of this review.

Line 499-509: Poorly written with repetition – rewrite and shorten.

Line 501: Delete “approximately”

Line 502: Rewrite: “not a typical form, being clinically variable”

Line 632-633: It might be worth explaining in more detail/ context the significance of “candidatus” species if you leave this information in, although I do not think this information is needed.

Line 659: But what level of seropositivity? I had to look up the reference. Almost 14% of samples seems relatively significant to me, there is little discussion of possible public health implications of this or whether further surveillance is warranted.

Line 704-717: For readers outside of Colombia, this long list of geographic areas is not that useful. I think overall the focus should be more on generalities – such as what geographic regions most of the cases are from- such as the north of the county, coastal regions, mountain regions, rural areas etc. This would give more useful information for the reader.

Line 713: “stablishing” typo

Figure 1: While somewhat useful these images are not that clear and need some editing.

This review needs extensive editing as I have many concerns with grammar, sentence structure and general readability. Repetition needs be be reduced and it could be much more concise, while containing the same information.

Reviewer 3 Report

The aim of this review paper by Silva-Ramos et al. is to argue for broadening the surveillance scope and provide awareness to possible causes of febrile illness in Columbia not currently covered by governmental surveillance programmes including bacterial, viral, protozoal and fungal causes that could pose a danger to Colombian public health. It brings attention to previous suspected and established cases. It additionally provides summaries for the symptoms, mortality and morbidity rates, epidemiology for the associated pathogens and possible animals that could be acting as viral/bacterial reservoirs. It supplies information about current diagnostic methods employed for detection of each pathogen in question including drawbacks and limitations. 

The review appears to be well-researched, comprehensive and appropriately referenced. The paragraphs describing individual diseases and their diagnosis are well written. The other sections of the manuscript need extensive proof reading. The introduction and conclusion sections should give more background information on what the current burden of febrile infections is in Colombia, which facilities are available for diagnostics of these infections and which methods are already established. The authors should speculate what the realistic public health impact of the additional diseases to be surveilled may be. How realistic is it, given financial, logistic and technical contrains that may exist, that these pathogens can be included in surveillance programmes? Following your in-depth analysis, what are the key risk factors for exposure to these pathogens in Colombia (insects, occupation, travelling etc.)? Which are the most affected areas? 

Major comments:

- The abstract, introduction and conclusion sections are very repetitive. One sentence to describe the different climate and vegetation zones and how this relates to disease prevalence would have been sufficient. The introduction should focus more clearly on where the pathogens in question arise from (wildlife, insects, travellers, livestock etc.) in relation to the above mentioned climate and vegetation zones. I thought this could have been written a lot more comprehensively and convincingly. 

- a table summarising the key information about the different pathogens to be included in surveillance would be useful

- the review is very extensive; I would suggest shortening wherever possible

Minor comments:

Lines 16 and 32: It is not clear what "diverse" refers to. 

Line 38. "‘experiencing climate change" rather than "experimenting"

Line 43-44: "being most of them" should be replaced with "however, most of them"

Line 44: You state that febrile illnesses are often zoonotic diseases (transmitted to humans from an animal reservoir), however, neither malaria, dengue, chikungunya, Zika, or typhoid fever are truely zoonotic unless you define zoonotic to include transmitted between people by insects. Indeed, these have an insect vector. This gives you an opportunity to discuss that the climate in Colombia is suitable for mosquitoes etc. to exist (see comment regarding climate above). 

Line 51: there is (not "are") scientific ... evidence that confirms or suggests

Line 52-53: Include further reasoning that would justify why these pathogens should be monitored as part of the Colombia government initiative on surveilling these pathogens. Why are these important? How damaging will the consequences be? Maybe draw comparisons to other countries that have experienced outbreaks with these pathogens.

Line 77: ‘After’ is repeated twice

Line 78: can develop

Line 81: "includes the detection or seroconversion of antibodies" should read "includes the detection of antibodies..."

Line 82. Should mention what IFA stands for – Indirect fluorescent antibody test (IFA)

Line 87: Do CL4 laboratories exist in Colombia?

Line 88-89. Should mention what species Oryzomys albigularis is (a rat species)

Line 97: delete "in both regions"

Lines 98-108: Given that there already is diagnostic evidence for GTOV circulation in Colombian rodents (line 106), I would lead with this and then speculate about human infections in the following sentences. 

Line 101. Perhaps should summarise Venezuelan hemorrhagic fever (GTOV?) case load in Venezuela

Line 103: neighboring 

Line 118: 22 of them being pathogenic

Line 128: Hantaan virus being the 

Line 143: needs

Line 161: In neither study (not "in none of both")

Line 197: causes

Line 208: "This last rarely used" - please correct the grammar

Line 226: stated, not "stayed"

Line 234: This evidence proves

Line 247: C. paraensis and C. quinquefasciatus being the vectors

Line 263: mention what ‘PRNT’ stands for

Line 270: no human cases

Line 312: … are also used for diagnosis

Line 318: whose

Lines 338/356/408: abbreviate Bartonella to B. when mentioned for the second etc. time 

Lines 346/361: delete "among others"

Line 373: comparable?

Lines 411-416: abbreviate Borrelia to B. when mentioned for the second etc. time

Line 439: needs

Line 470: This data must be

Line 471: Reference missing for the cross-reactivity of Borrelia species; replace "being" by "It is" (new sentence)

Line 487: ..historical data to state that 

Line 490-491: Mentioning of staining used for Coxiella should be moved to the diagnostic part of the paragraph, and state how or why it is useful

Line 501: it does not have a typical form

Line 509: the last two being 

Line 511: What is meant by phase I and phase II in C. burnetii infections?

Line 520: replace "was done" by "was detected" or "occurred"

Lines 525/534: who may have acquired the infection

Line 537: of Colombia.

Line 569 onwards: abbreviate Ehrlichia to E. when mentioned for the second etc. time

Line 656: "though" change to "thought"

Line 670-674: abbreviate Rickettsia to R. when mentioned for the second etc. time

Line 681: R. rickettsii being the deadliest one

Line 732: have been reported

Line 749: the first one being

Line 782: change "even some of them remain unaware" to "and some of them even remain undiscovered"

Line 788: replace "and those of which there are already enough evidence of" with "and those for which there is already enough evidence of"

Figure 1: What is the relevance of the numbers within the coloured dots? There seem to be more cases of febrile illnesses in the North-West of the country. What is the reason for that?

I pointed out a number of sentences where the English needs to be corrected, however there were certainly more. 

Round 2

Reviewer 2 Report

The authors have substantially improved the manuscript after the first reviews and it reads much better. I also think the focus of the whole article is improved and some repetitive or less relevant parts have been taken out.

The addition of the seroprevalence tables is good.

Overall the authors have done a good job of responding to reviewer comments. Overall it is an interesting and informative read and will be off benefit to help consider the approach for public health surveillance systems in Colombia.

I have no major concerns but would comment that there still needs some minor review of the writing. I have not given comprehensive examples but some things like these below give an idea of minor issues still:

CDC (Center for Disease Control and Prevention)- “Centers…”

“for which few are known to date regarding its eco-epidemiology, so further studies" Few of what? Suggest re-word.

While the added parts in the conclusion are useful and help define the scope / argument of the article better, I also think this could be better written.

I would suggest the tracked changes are tidied up and then a further few reading and edits are made to tidy up the text and eliminate any remaining errors.

Reviewer 3 Report

Dear Authors, dear Editor,

I appreciate the efforts made in improving the manuscript. The newly added tables are indeed very useful. Given that the manuscript is overall long, and readability is thus very important, I would like to suggest the following minor changes, especially to the language used, before publication. As you may have noticed, the manuscript already reads so much better, strengthening your message rather than distracting from it. So thank you for bearing with me. 

Minor comments:

- I suggest to rewrite the abstract as follows:

In Colombia, tropical febrile illnesses represent one of the most important causes of clinical attention. They are mainly zoonotic and have a broad etiology. The Colombian surveillance system monitors several compulsory-reporting diseases, however, surveillance is not comprehensive. In the present review, we describe eleven different etiologies of zoonotic tropical febrile illnesses for which scientific, historical, and contemporary data exists, which confirms or suggests their presence in different regions of the country, but which are not currently monitored. These include: Anaplasma, Arenavirus, Bartonella, Borrelia, Coxiella burnetii, Ehrlichia, Hantavirus, Mayaro virus, Orientia, Oropouche virus, and Rickettsia. They could pose a significant health risk to the local population, travelers, and immigrants, and should thus be included in the mandatory notification system. 

- line 57: several others, whose importance has been demonstrated in other countries

- line 62: in Colombia.

- lines 63-71: can you please suggest a pipeline for how these samples would be analysed. Are you suggesting that samples will be collected locally and then sent abroad for analysis? Is this realistic?

- line 76: have traditionally been divided

- line 92: resolves

- line 101: of IgM or IgG antibodies

- line 107: however, the last method is 

- line 118: among different population groups (Table 1) (refs)

- line 124: that another arenavirus may be present in Colombia

- line 128:  in Zygodontomys brevicauda rodents from rural

- line 152: 22 of which being pathogenic

- lines 158-171: These two sentences are too long. Can you please split them up into shorter sentences for improved readability. 

- lines 199-203: Can you please correct this sentence and start a new sentence at "In neither study, ".

- line 212: in whom

- line 222: whose importance

- line 224: for which little is known to date. Further studies among wild rodent populations are thus needed because

- line 249: half of the patients

- line 272: the highest being

- line 295: whose competence

- line 335: this being

- line 348: the reference ought to be at the end of the sentence

- line 356: transmitted by tick bites

- line 360: novel

- line 390: and who fully recovered

- line 391: delete "also"

- line 395: supporting the hypothesis

- line 429: the chronic phase of the disease

- line 491: in clothing lice

- line 497: could be circulating

- line 519: the genus Ornithodorus spp. being

- line 560: reference ought to be at the end of the sentence

- line 590: two antigenic phase variants?

- line 593: acute forms of the disease

- line 599: with contact to cattle

- line 609: the latter two

- lines 619/620: please correct this sentence

- Lines 630-635: please split this sentence into two or three rather than using multiple semicolons

- line 651: please start a new sentence

- line 661: the reference is to be placed at the end of the sentence

- lines 682/683: start a new sentence after "hosts" and delete "being"

- line 694: please start a new sentence at "However,"

- line 722: with contact to

- line 735: reference to go to the end of the sentence

- line 749: both being pathogenic

- line 763: start a new sentence with "In some cases..."

- line 819: delete "highly"

- line 826: What does "they are" refer to?

- line 863: delete "Furthermore,"

- line 885: may also be considered an important

- line 895: should alert local authorities

- line 898: five of them being coinfections

- line 922: please end the sentence after "unknown"

- line 922: "It is thus necessary to carry out more studies using the One Health approach, and to understand the importance..."

- line 925: please end the sentence after "territory"

- line 928 etc.: suggesting their inclusion into the mandatory notification system. - delete the rest 

Figure 1: I asked what the reason is for most of the diseases being more prevalent in the NW of Colombia and you said these diseases have not been studied in detail elsewhere. Would it make sense to mention this limitation  somewhere in the tex, maybe in the figure legend, or have an extra sub figure here to show where investigations have and have not been undertaken. Otherwise, it may be misleading and readers may think diseases are restricted to the NW for ecologic etc. reasons. 

Dear Authors, dear Editor, 

This manuscript has certainly improved significantly. There need to be a few more changes to the language made, please. I understand a formal English review will also be undertaken. 
